# Serum procalcitonin level is independently associated with mechanical ventilation and case-fatality in hospitalized COVID-19-positive US veterans–A potential marker for disease severity

Sujee Jeyapalina[1,2☯]*, Guo Wei[2], Gregory J. Stoddard[2], Jack D. Sudduth[1], Margaret Lundquist[3], Merodean Huntsman[1,2], Jessica L. Marquez[1], Jayant P. Agarwal[1,2☯]*

1 Division of Plastic and Reconstructive Surgery, Department of Surgery, University of Utah School of Medicine, Salt Lake City, UT, United States of America, 2 Division of Epidemiology, Department of Internal Medicine, University of Utah School of Medicine, Salt Lake City, UT, United States of America, 3 Research, George E. Wahlen Department of Veterans Affairs Medical Center, Salt Lake City, UT, United States of America

☯ These authors contributed equally to this work.
* jay.agarwal@hsc.utah.edu (JPA); sujee.jeyapalina@hsc.utah.edu (SJ)

## Abstract

The Coronavirus-19 disease (COVID-19) has claimed over 6.8 million lives since first being reported in late 2019. The virus that causes COVID-19 disease is highly contagious and spreads rapidly. To date, there are no approved prognostic tools that could predict why some patients develop severe or fatal disease outcomes. Early COVID-19 studies found an association between procalcitonin (PCT) and hospitalization or duration of mechanical ventilation and death but were limited by the cohort sizes. Therefore, this study was designed to confirm the associations of PCT with COVID-19 disease severity outcomes in a large cohort. For this retrospective data analysis study, 27,154 COVID-19-positive US veterans with post-infection PCT laboratory test data and their disease severity outcomes were accessed using the VA electronic healthcare data. Cox regression models were used to test the association between serum PCT levels and disease outcomes while controlling for demographics and relevant confounding variables. The models demonstrated increasing disease severity (ventilation and death) with increasing PCT levels. For PCT serum levels above 0.20 ng/ml, the unadjusted risk increased nearly 2.3-fold for mechanical ventilation (hazard ratio, HR, 2.26, 95%CI: 2.11–2.42) and in-hospital death (HR, 2.28, 95%CI: 2.16–2.41). Even when adjusted for demographics, diabetes, pneumonia, antibiotic use, white blood cell count, and serum C-reactive protein levels, the risks remained relatively high for mechanical ventilation (HR, 1.80, 95%CI: 1.67–1.94) and death (HR, 1.76, 95%CI: 1.66–1.87). These data suggest that higher PCT levels have independent associations with ventilation and in-hospital death in veterans with COVID-19 disease, validating previous findings. The data suggested that serum PCT

**Data Availability Statement:** All relevant data are within the paper and its Supporting Information files.

**Funding:** This investigation was supported by an unrestricted investigator-initiated research grant from Gilead Sciences (# CO-US-983-6072). The research was also, in part, supported by the National Center for Advancing Translational Sciences of the National Institutes of Health under Award Number UL1TR002538. The content is solely the responsibility of the authors and does not necessarily represent the official views of the National Institutes of Health or Gilead Scientific Inc. The funders had no role in study design, data collection, and analysis, decision to publish, or preparation of the manuscript.

**Competing interests:** The authors have declared that no competing interests exist.

level may be a promising prognostic tool for COVID-19 severity assessment and should be further evaluated in a prospective clinical trial.

## Introduction

The coronavirus-19 disease (COVID-19) is a novel illness caused by severe acute respiratory syndrome coronavirus 2 (SARS-CoV-2). This virus was originally diagnosed in a cluster of patients in December 2019, the so-called "mysterious pneumonia cases" of unknown etiology in China [1–3]. Even after 24 months, this human-to-human transmittable virus continues to claim many lives. As of the 15th of March 2023, the World Health Organization (WHO) reported a staggering ~6.8 million deaths worldwide [4]. It is worth noting that this number may be an underestimation owing to some countries under-reporting their actual case casualties [5, 6]. Although understanding the spread dynamics of coronaviruses can help to project and also curb the outbreak by using public health measures, to date, there is no patient-specific prognostic laboratory tool/test that would predict why some suffer from severe disease symptoms and die from this disease while others remain asymptomatic. A reliable prognostic test might help triage patients to an appropriate level of care within the hospitals.

To date, there has been very little progress made with prognostic tests for the early reliable identification of worsening disease outcomes. Most of the studies discussed clinical features and radiological findings [7–13], whereas only a handful of studies reported the prognostic value of abnormal laboratory findings [14–26]. Amongst them, serum level markers, as well as immune cell counts, have all been associated with symptom severities, including case-fatalities [14–26]. Although they could be valuable prognostic tools, most of these blood markers have known associations with systemic inflammation and bacterial or viral diseases. They may not be specific for predicting worsening COVID-19 symptom severity.

Previous COVID-19 studies have suggested that the blood plasma laboratory values of PCT as well as others such as CRP, D-dimer, etc., are often elevated in patients with COVID-19 who required ICU admission, mechanical ventilation (MV), or those who died [23, 27–37]. Many COVID-19 studies reported elevated PCT levels in hospitalized patients or those placed on mechanical ventilators [30, 32, 34, 35]. One meta-analysis reported that increased procalcitonin values were associated with a nearly 5-fold higher risk of severe SARS-CoV-2 infection [29]. In a second meta-analysis study, Ahmed *et al.* concluded that 85% (n = 44) of the studies reported a statistically significant association between elevated PCT and disease severity [23]. Based on the literature, there appears to be a strong association between high PCT serum levels amongst COVID-19 patients with severe symptoms; thus, serum PCT tests may warrant further consideration as an early predictor. Additionally, it is unknown whether PCT levels can be tracked serially to show a trend in patient condition. Historically, PCT level has been mainly associated with bacterial sepsis [38–49] and not with viral infections [50–53]. However, a few studies have shown elevated PCT levels in patients with multi-system organ failure and systemic inflammation without bacterial infection [54–57]. Regardless of the direct cause of the elevated PCT level, there appears to be an association between the level of PCT in patients with severe COVID-19 symptoms [23, 30–33, 37, 58–62]. Many of these studies had smaller cohorts; thus, there is a need to further evaluate the possible association between serum PCT levels and COVID-19 disease outcomes.

This study was therefore designed to test the primary hypothesis that there would be an association between elevated serum PCT values and symptom severity in a larger cohort of hospitalized Veterans with COVID-19 who were admitted to the nationwide Veterans Health

Administration hospital system. If the association is replicated in this study, it would further support the pertinent benefit of using PCT to guide the prognosis of worsening COVID-19 disease with further retrospective studies.

## Materials and methods

The Institutional Review Boards (IRB) of the University of Utah and the Department of Veterans Affairs Salt Lake City Hospital system reviewed this retrospective study protocol and deemed it exempted, and waived the requirement for informed consent. The ethical approvals (IRB# 00133238) were received on June 23, 2020. The COVID-19 Shared Data Resources were accessed and analyzed using the VA informatics and computing infrastructure server (VINCI) post-approval. This way, veterans' privacy and data security were ensured. This shared data resource collects information about veterans who were diagnosed with COVID-19 within VHA from the VHA's Corporate Data Warehouse (CDW) [63]. The CDW was also accessed to get longitudinal PCT laboratory values. Patients' personal information was deidentified prior to this retrospective data analysis.

### Study design and data resources

This retrospective study used data from the nationwide Department of Veterans Hospital Administration (VHA) electronic medical records of hospitalized US veterans with laboratory-confirmed positivity for COVID-19 within 7-days after the index date. The index date was defined within the VA's Shared COVID-19 Data Resources as the first positive COVID-19 test or the inpatient admission date if veterans received care within 15 days before the positive test. All veterans admitted to the VHA hospital system between March 1, 2020, and February 28, 2022, with 60-day outcomes for in-hospital death or discontinued ventilator use (i.e., up to 27th April 2022) were pooled for this study. Additional inclusion criteria include veterans over 18 years of age and with at least one serum PCT test data. A selection of other information such as index date, death or hospital discharge date, comorbidities, presence or absence of pneumonia, antibiotic use, and lab values for inflammatory blood markers were obtained.

### Outcomes

Mechanical ventilator use or death.

### Key study variable

The variable of interest in this cohort study was the presence of elevated serum PCT, C-reactive protein, lactate, and white blood cell count. The use of antibiotics was considered as the surrogate marker for pneumonia in the absence of bacterial culture data. The first available procalcitonin level during hospitalization was chosen as a predictor variable.

### Covariates

To identify comorbidities and patient characteristics predictive of severe disease outcomes, the following predictor variables were extracted from the data resources as potential confounders: age, race, obesity, history of diabetes mellitus, heart failures, chronic kidney disease (CKD), and chronic liver disease.

### Statistical analysis

The study hypothesis was that serum PCT level would be associated with the clinical course of patients admitted to the hospital with a COVID-19 infection. Recognizing that mechanical

ventilation is an intermediate therapy preceding death or discharge in the clinical course, we report serum PCT level and demographic variables using four clinical course categories: recovered (discharged) without ventilation, recovered after ventilation, fatality without ventilation, and fatality after ventilation. For the total cohort and the four clinical course categories, we showed demographic variables using counts and percentages in each of the four clinical course categories. For serum PCT level, a continuous variable with a right-skewed distribution, we show the median and interquartile range (IQR: 25th and 75th percentiles).

Using a 60-day follow-up period beginning with hospital admission, mechanical ventilation, and death were treated as separate binary outcomes. To model these two outcomes, serum PCT was dichotomized into a low and high group (low: ≤0.2 ng/ml; high: >0.2 ng/ml). During the 60-day follow-up period, given that patients could have crossover between serum PCT categories and other covariate values could have changed, we used time-varying Cox regression models and time-varying Kaplan-Meier plots. In particular, all available longitudinal PCT tests for each patient were included in the models in a time-varying fashion. The primary predictor variable was binary serum PCT, using low as the referent category. The models adjusted for demographics (age, race), comorbid conditions (obesity, diabetes, CKD, hypertension, and liver disease), antibiotic status (pneumonia and use of antibiotics), and lab results (CRP, lactate, and white blood cell count).

To further confirm the serum PCT association with outcomes, using the subgroup of patients with at least two serum PCT tests, we created a binary serum PCT change variable between the first and last serum PCT tests, scored as 1, if decreased (clinically desirable result), versus 0 if increased or no change (clinically undesirable result). Using increase as the referent category, a hazard ratio < 1 would suggest a protective association. We compared the four outcome subgroups on this variable using a chi-square test. Additionally, to illustrate how much change occurred in the patient cohort, we graphed the continuous serum PCT level using boxplots, displaying the first and last test values for each of the four clinical course categories and comparing the first and last values with a paired sample Wilcoxon signed-rank test. All reported p values are for a two-sided comparison. All statistical analyses were performed using Stata Version 15.1 (College Station, TX, USA).

## Results

Fig 1 shows that there were 581,884 veterans reported to have had COVID-19 infection during this retrospective study period. Of those, 74,195 (12.7%) were hospitalized and treated within the VHA hospital system. Since PCT levels were not obtained in all patients, only those who had records of one or more serum PCT test data were included in this retrospective data analysis study, which resulted in a pool of 27,157 (36.5%) hospitalized veterans. Table 1 shows the demographic characteristics of this cohort. Among this cohort, nearly 67.4% were 65 years or older, and 63.2% were of the White race. The majority (75.2%) had a BMI of 25 or over. The median and interquartile range (IQR) of PCT level within the hospitalized but recovered and discharged veterans without an event was ~ 0.1(0.1–0.3) ng/ml. This value was 3 times higher in veterans who suffered in-hospital death (0.3 (0.1–0.8) ng/ml) after receiving mechanical ventilation and 2 times higher for those who recovered after ventilation or those who perished without ventilation (0.2 (0.1–0.6) ng/ml). Table 2 shows the trend of serum PCL changes in patients who had multiple measurements.

Table 3 shows the prevalence of pneumonia, antibiotic use, and pre-index comorbidities among the studied population. This information demonstrates that, although 22,606 (~83%) veterans' charts had a diagnosis code for pneumonia, 17,763 (~65%) veterans had a record of antibiotic use after the COVID-19 diagnosis. In the absence of blood culture data, antibiotics

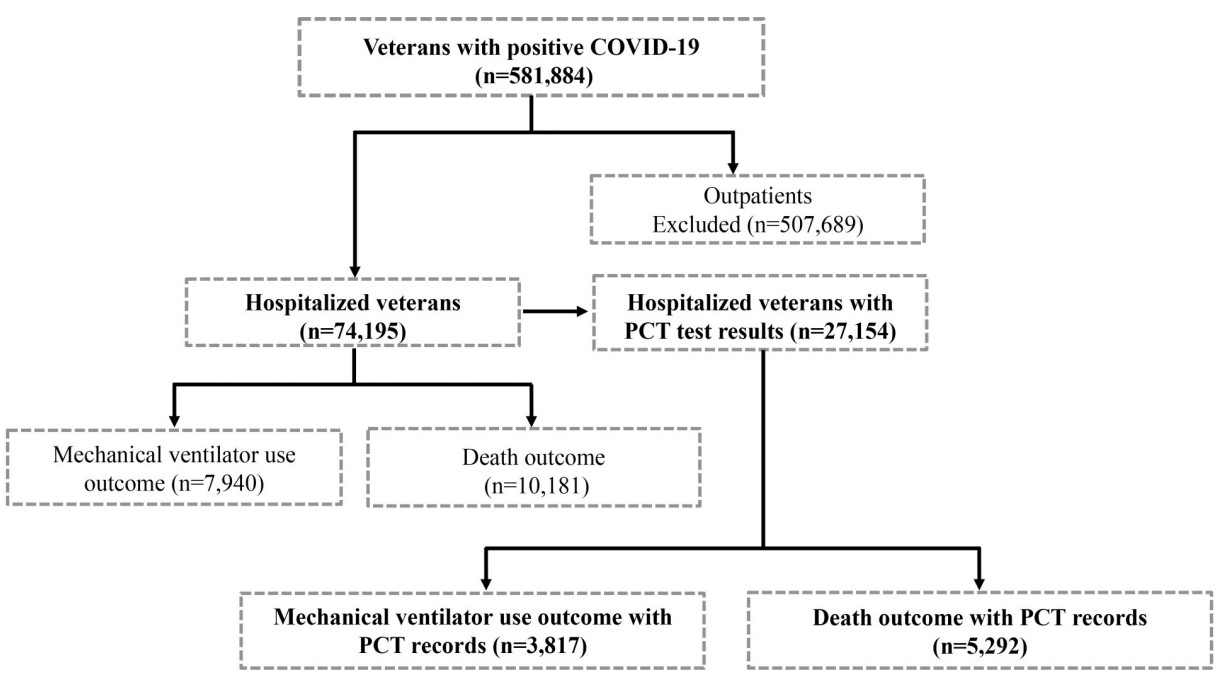

**Fig 1. A flow chart showing the reported COVID-19-positive cases amongst veterans with and without the records of serum PCT test results between March 1, 2020, and February 28, 2022.**

were considered a surrogate for bacterial co-infections. Among recovered veterans without mechanical ventilation use, 39.5% (n = 8,117) did not receive antibiotics. Severity appeared to vary by comorbidity, with Liver disease (90.7%) and HTN (82.0%) being the most prevalent diseases in the group, with in-hospital death as the outcome. The average PCT levels (0.3 (0.1–0.8) ng/ml) were highest in patients who were mechanically ventilated (MV) and then suffered in-hospital death. There were a number of other differences in laboratory findings between the elevated PCT group and the normal PCT group, which included CRP, lactate, as well as WBC, which is given in S1 Table.

Among these 27,154 hospitalized veterans included in the study, 9,096 (33.5%) veterans had ≤0.1ng/ml serum PCT level, which included 2,179 veterans with normal levels (<0.05ng/ml). Approximately 86% of them (n = 7,832) were discharged from the hospital without MV. The remaining 14% either received mechanical ventilation or succumbed to in-hospital death. Out of this hospitalized patient pool, 10,813 veterans had multiple PCT evaluations during their hospital stay, and their first and last PCT levels are graphically depicted in Fig 2. Interestingly, the first and last test median values for those who were not mechanically ventilated and discharged from the hospital had median value decreased from ~0.16 to 0.13ng/ml, while veterans who received mechanical ventilation or suffered in-hospital death had increased median PCT values (Fig 2). There were statistically significant changes between the first and last serum PCT values in each of the four subgroups (Wilcoxon signed-rank test, all p<0.05) (Fig 2). It is worth noting that the hospitalized and then discharged groups had decreasing trajectories (Fig 2). Although the distribution of PCT data is positively skewed (skewed right, Fig 2), long right tails on the distribution of mechanically ventilated and in-hospital mortality groups indicate that PCT levels have a wide range of distributions. Also, the initial univariable analysis confirmed that patients with an initial PCT level of >0.2 ng/ml had the worst disease outcomes (i.e., death or mechanical ventilation). Thus, this 0.2 ng/ml value was considered appropriate

**Table 1. Characteristics of 27,154 COVID-19-positive US veterans with at least one PCT measurement documented in their medical records between 01 March 2021 and 28 February 2022.** *Unknown in the race, age, and BMI categories are those without any entry in the respective fields.

| | Hospitalized (n) | Recovered without ventilation | Recovered after ventilation | Fatality without ventilation | Fatality after ventilation | p-value |
|---|---|---|---|---|---|---|
| **No: of COVID-19 positive veterans with serum PCT data** | 27,154 | 20,531 | 1,331 | 2,806 | 2,486 | |
| PCT level (ng/ml) (median (IQR) | 0.2 (0.1–0.4) | 0.1 (0.1–0.3) | 0.2 (0.1–0.6) | 0.2 (0.1–0.6) | 0.3 (0.1–0.8) | <0.001 |
| **0.1≥PCT (ng/ml)** | | | | | | <0.001 |
| Reported | 9,098 (33.5%) | 7,832 (38.1%) | 284 (21.3%) | 607 (21.6%) | 375 (15.1%) | |
| **Age groups** | | | | | | <0.001 |
| <50 | 2,520 (9.3%) | 2,263 (11.0%) | 163 (12.2%) | 9 (0.3%) | 85 (3.4%) | |
| 50 - <65 | 6,322 (23.3%) | 5,306 (25.8%) | 378 (28.4%) | 160 (5.7%) | 478 (19.2%) | |
| > = 65 | 18,312 (67.4%) | 12,962 (63.1%) | 790 (59.4%) | 2,637 (94.0%) | 1,923 (77.4%) | |
| Unknown | 2,520 (9.3%) | 2,263 (11.0%) | 163 (12.2%) | 9 (0.3%) | 85 (3.4%) | |
| **BMI groups(kg/m$^2$)** | | | | | | <0.001 |
| Underweight (< 18.5) | 814 (3.0%) | 535 (2.6%) | 29 (2.2%) | 203 (7.2%) | 47 (1.9%) | |
| Normal weight (18.5–24.9) | 5,925 (21.8%) | 4,279 (20.8%) | 190 (14.3%) | 1,011 (36.0%) | 445 (17.9%) | |
| Overweight (25–29.9) | 8,139 (30.0%) | 6,161 (30.0%) | 363 (27.3%) | 866 (30.9%) | 749 (30.1%) | |
| Obese (30–39.9) | 9,957 (36.7%) | 7,769 (37.8%) | 595 (44.7%) | 603 (21.5%) | 990 (39.8%) | |
| Morbidly Obese (40+) | 2,238 (8.2%) | 1,737 (8.5%) | 153 (11.5%) | 99 (3.5%) | 249 (10.0%) | |
| *Unknown | 81 (0.3%) | 50 (0.2%) | 1 (0.1%) | 24 (0.9%) | 6 (0.2%) | |
| **Race** | | | | | | <0.001 |
| American Indian or Alaska Native | 229 (0.8%) | 159 (0.8%) | 20 (1.5%) | 22 (0.8%) | 28 (1.1%) | |
| Asian | 225 (0.8%) | 171 (0.8%) | 14 (1.1%) | 18 (0.6%) | 22 (0.9%) | |
| Black or African American | 7,341 (27.0%) | 5,741 (28.0%) | 375 (28.2%) | 549 (19.6%) | 676 (27.2%) | |
| Native Hawaiian or Other Pacific Islander | 263 (1.0%) | 201 (1.0%) | 14 (1.1%) | 27 (1.0%) | 21 (0.8%) | |
| White | 17,159 (63.2%) | 12,796 (62.3%) | 808 (60.7%) | 1,987 (70.8%) | 1,568 (63.1%) | |
| *Unknow | 1,937 (7.1%) | 1,463 (7.1%) | 100 (7.5%) | 203 (7.2%) | 171 (6.9%) | |
| **Selected comorbidities (2 years pre-ID)** | | | | | | |
| HTN | 20,816 (76.7%) | 15,437 (75.2%) | 1,040 (78.1%) | 2,315 (82.5%) | 2,024 (81.4%) | <0.001 |
| Diabetes | 13,070 (48.1%) | 9,636 (46.9%) | 692 (52.0%) | 1,374 (49.0%) | 1,368 (55.0%) | <0.001 |
| CKD | 7,606 (28.0%) | 5,382 (26.2%) | 338 (25.4%) | 1,075 (38.3%) | 811 (32.6%) | <0.001 |
| Heart Failure | 5,768 (21.2%) | 3,987 (19.4%) | 288 (21.6%) | 877 (31.3%) | 616 (24.8%) | <0.001 |
| Liver disease | 2,582 (9.5%) | 1,944 (9.5%) | 145 (10.9%) | 236 (8.4%) | 257 (10.3%) | 0.03 |

**Table 2. Trend trajectories showing the changes in the serum PCT values and recorded COVID-19 severity outcomes of hospitalized veterans (n = 11,833; 43.5% of the study cohort) with two or more serial measurements.**

| | Hospitalized | Recovered without ventilation | Recovered with ventilation | Fatality without ventilation | Fatality with ventilation | P value |
|---|---|---|---|---|---|---|
| PCT change (The last vs. first measurement before death for patients with 2 or more measurements, n = 11,833;), which is a larger sample size than MV (shown 3 lines lower), with longer follow-up time death, there was more opportunity to have multiple PCT measurements. | | | | | | <0.001 |
| PCT increased | 4,738 (40.0%) | 2,353 (30.5%) | 411 (44.3%) | 720 (51.5%) | 1,254 (69.7%) | |
| PCT no change | 602 (5.1%) | 510 (6.6%) | 18 (1.9%) | 52 (3.7%) | 22 (1.2%) | |
| PCT decreased | 6,493 (54.9%) | 4,846 (62.9%) | 498 (53.7%) | 626 (44.8%) | 523 (29.1%) | |
| PCT change (The last vs. first measurement before MV for patients with 2 or more measurements, n = 10,813;) | | | | | | <0.001 |
| PCT increased | 3,944 (36.5%) | 2,354 (30.5%) | 230 (45.9%) | 720 (51.5%) | 640 (53.2%) | |
| PCT no change | 609 (5.6%) | 508 (6.6%) | 20 (4.0%) | 52 (3.7%) | 29 (2.4%) | |
| PCT decreased | 6,260 (57.9%) | 4,847 (62.9%) | 251 (50.1%) | 627 (44.8%) | 535 (44.4%) | |

**Table 3. Reported cases of pneumonia and antibiotic use within the study cohort of 27,154 COVID-19-positive US veterans.**

| | Total Hospitalized | Recovered without ventilation | Recovered after ventilation | Fatality without ventilation | Fatality after ventilation | p-value |
|---|---|---|---|---|---|---|
| Number of veterans (n) | 27,154 | 20,531 | 1,331 | 2,806 | 2,486 | |
| PCT (ng/ml)(IQR) | 0.2 (0.1–0.4) | 0.1 (0.1–0.3) | 0.2 (0.1–0.6) | 0.2 (0.1–0.6) | 0.3 (0.1–0.8) | <0.001 |
| **PCT level** | | | | | | |
| PCT < = 0.1 | 9,098 (33.5%) | 7,832 (38.1%) | 284 (21.3%) | 607 (21.6%) | 375 (15.1%) | <0.001 |
| PCT < = 0.2 | 16,138 (59.4%) | 13,332 (64.9%) | 611 (45.9%) | 1,280 (45.6%) | 915 (36.8%) | <0.001 |
| Pneumonia (reported within 60 days post-ID) | 22,606 (83.3%) | 16,476 (80.2%) | 1,227 (92.2%) | 2,476 (88.2%) | 2,427 (97.6%) | <0.001 |
| Antibiotic use (reported within 60 days post-ID) | 17,763 (65.4%) | 12,414 (60.5%) | 1,148 (86.3%) | 2,045 (72.9%) | 2,156 (86.7%) | <0.001 |

as the cut-off value for Cox regression analyses. S2 Table further confirms that veterans with decreasing PCT values during hospitalization had better outcomes.

For the Cox regression analyses, models with an increasing number of covariates were used (Fig 3): Model 1—unadjusted and three adjusted Cox Regression models. Multivariable Cox regression models (models 1–4, Fig 3) clearly showed the risk of mechanical ventilation and in-hospital death for those with higher than 0.2ng/ml PCT levels. As seen in Fig 3, unadjusted and adjusted for age, BMI, race, sex, history of CKD, HTN, liver disease, diabetes, reported pneumonia, antibiotic use, and blood CRP, WBC, as well as lactate levels did not weaken the risk associations. Overall, the data revealed an approximately 2-fold increase in risk for mechanical ventilator use and in-hospital death in patients with a PCT level of 0.2ng/ml or higher. The unadjusted HRs were only slightly reduced in the adjusted models (Fig 3). Moreover, the Kaplan-Meier curves illustrate significantly better disease outcomes for patients with lower serum PCT values (Fig 4).

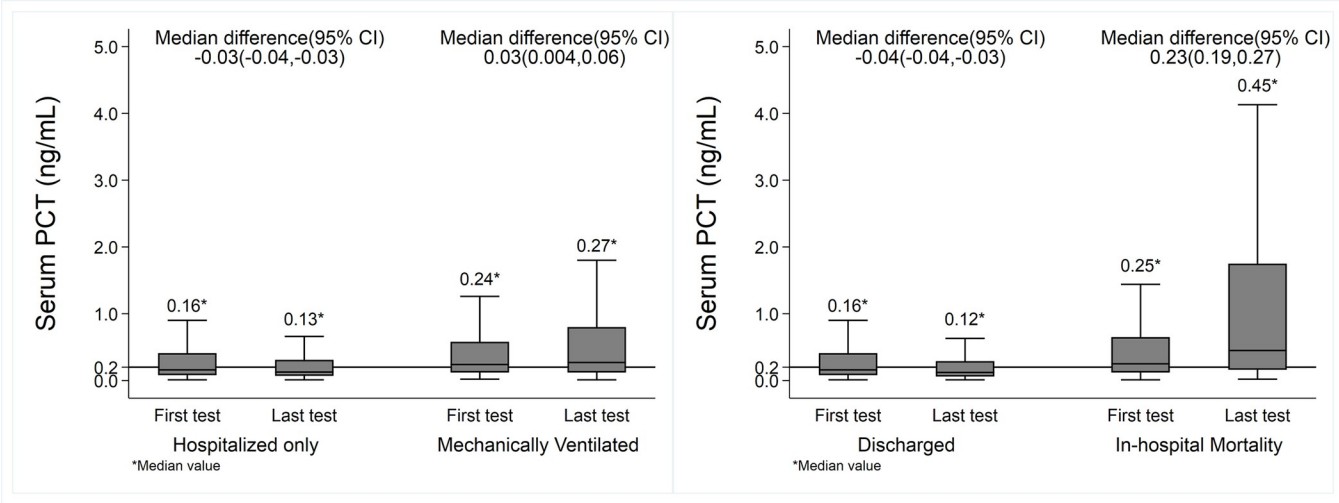

**Fig 2. Boxplots showing a 5-number summary (minimum, first quartile, median, third quartile, and maximum) for first and last serum PCT levels and stratified based on final outcomes.** The majority of the first test values were obtained within 2–16 days of hospitalization, while the last value was immediately prior to the reported outcomes of hospital discharge, mechanical ventilation, or in-hospital death. The boxes extend from the 25th to the 75th percentile, with whiskers extending to the minimum and maximum points. Outliers are not shown in the box plot.

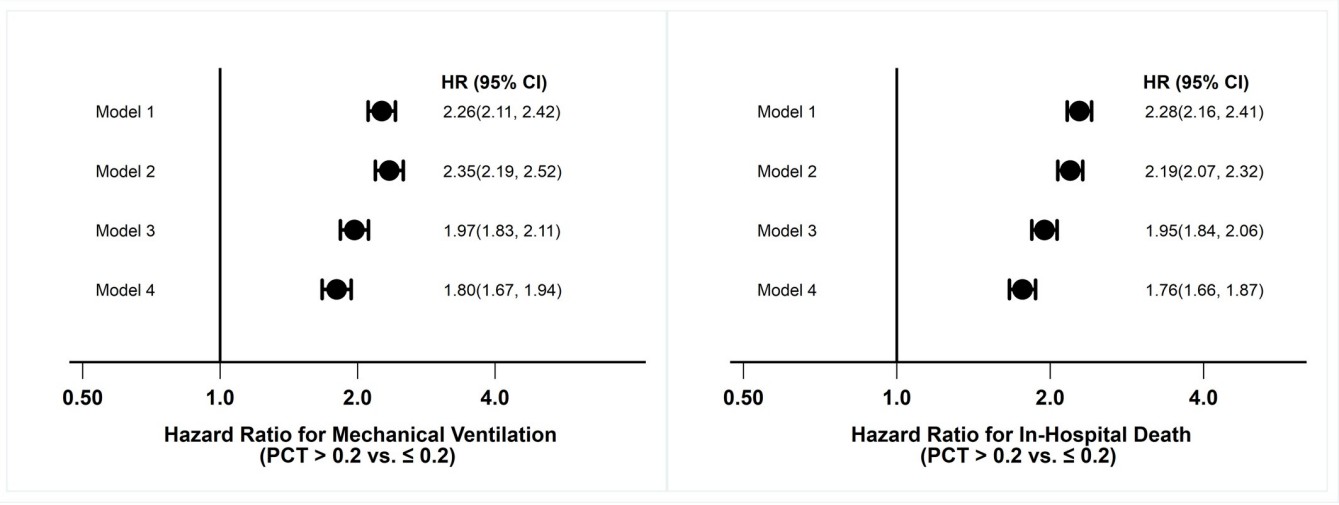

**Fig 3. Forest plots showing the hazard ratios for mechanical ventilation and in-hospital death.** Model 1 is unadjusted. Model 2 adjusted for demographic and comorbidity conditions (i.e., adjusted for age, race, BMI, history of CKD, Heart failure, hypertension, liver disease, and diabetes), Model 3 adjusted for demography, comorbidity conditions (Model 2) as well as pneumonia (presence or absence) and antibiotic use (used or not used). Model 4 adjusted for all covariates of Model 3 and lab test results (CRP, WBC, and lactate).

## Discussion

Overall, multivariable Cox regression models of COVID-19-positive veterans with records of serum PCT values support that high serum values were positively associated with disease progression and severity for MV (adjusted HR, 1.80, 95%CI: 1.67–1.94) and in-hospital death (adjusted HR, 1.76, 95%CI: 1.66–1.87) (Fig 3). These models also showed worsening severity during MV for those with high serum PCT values, supporting the tested hypothesis. There were also differences in PCT trend trajectories and overall severity outcomes in the patients

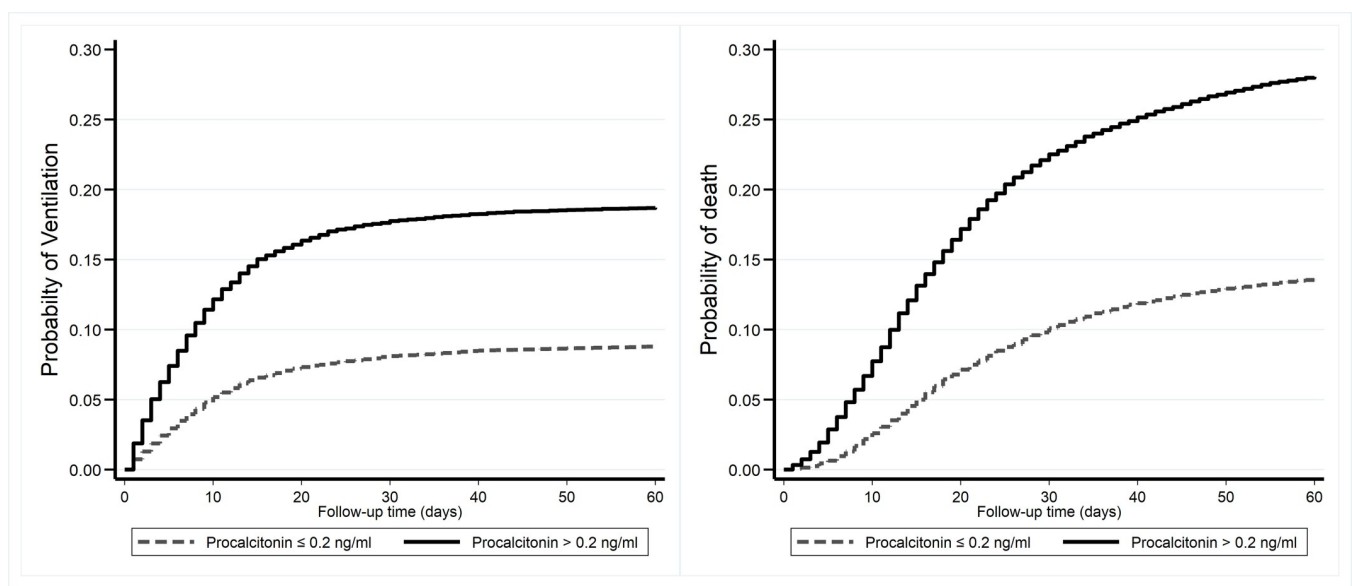

**Fig 4.** Kaplan–Meier plots for the probability of mechanical ventilation use (A) survival (B) for PCT > 0.2 ng/ml (solid black lines) and PCT ≤ 0.2ng/ml (broken black lines).

with serial, longitudinal PCT data, suggesting longitudinal PCT levels could be used as a prognosticator for COVID-19 disease severity (S2 Table). The Kaplan-Meier curves (Fig 4) also supported the Cox regression models (Fig 3). Overall, these results show significant associations of higher serum PCT concentrations with both MV and in-hospital fatality.

COVID-19 patients with weakened immunity are also particularly vulnerable to bacterial and fungal superinfections [64–70]. Also, bacterial infections can result from the use of mechanical ventilators, which are commonly used to treat severe respiratory failure in patients with COVID-19 [65, 71, 72]. In the absence of a known link between PCT levels and viral infection, there is a possibility that the elevated PCT levels could be attributed to either bacterial co-infections [30, 38, 58, 73], multiorgan failure, or acute kidney injury [74]. Whether or not the elevated PCT levels are due to bacterial co-infection warrants further prospective studies. One COVID-19 study pointed out that patients with secondary bacterial infection had higher PCT and CRP levels when compared to the non-bacterial co-infection COVID-19 control group, concluding that these markers show high sensitivity and specificity in combination as a predicator for bacterial infection [73, 75], which is supported by other general sepsis studies [47, 53, 76–78]. In our cohort, 83.3% of the veterans had reported cases of pneumonia (Table 3). In the absence of culture results and other supporting data, it is difficult to discriminate between bacterial and viral pneumonia. From this database, it is impossible to determine why the antibiotics were prescribed. Thus, other laboratory findings, such as CRP levels, erythrocyte sedimentation rate (ESR), and elevated lactate dehydrogenase levels, were used to establish a correlation as surrogate variables to discriminate between bacterial from viral pneumonia. However, there were no significant correlations found. Since 4,548 (~16.7%) veterans did not have pneumonia while 9,391 (~34.6%) also did not receive any antibiotic treatments, it appears that antibiotics might have been given as the prophylactic treatment. It is also probable that some patients could have been only treated with empiric antibiotics if any bacterial infection was suspected at the point of care.

Interestingly, one study investigating the role of PCT results in antibiotic decision-making concluded that PCT levels might be abnormal in COVID-19 patients without bacterial pneumonia [59], suggesting bacterial co-infection may not be the only cause for elevated PCT levels. This study was further supported by Heer *et al*., who have shown that there was an association between PCT level and neutrophil count [79]. Although the elevated neutrophil number is often associated with bacterial infection, it is also reported to be elevated in patients with COVID-19-associated thrombosis [80, 81]. Thus, the elevated levels of PCT could also be indicative of blood clots and related-multiorgan failure, which is a common theme in severe COVID-19 cases [82–85]. Other organ failure literature supports this assumption, demonstrating an association between multiorgan failure and high serum PCT levels in severe viral infections [55–57, 86]. However, in multiorgan failure cases, the serum PCT levels are not as high as the sepsis and septic shock cases (median value of over 5.0 ng/ml) [76, 87]. It is worth noting that the average serum PCT level was 0.3 (0.1–0.8) ng/ml in our in-hospital fatality cases. To date, the PCT levels have yet to be associated with COVID-19-related multiorgan failure.

Most literature supports that PCT is not usually elevated with viral infections [50, 51, 88, 89]. It may be worthwhile to understand how PCT levels are controlled in the human body to grasp the relevance of COVID-19 severe symptoms assessment. PCT is the precursor protein of the hormone calcitonin, which controls the blood calcium level. In healthy individuals, PCT is readily converted to calcitonin; as such, the serum PCT values in non-infected, healthy individuals normally remain very low, i.e., < 0.05 ng/ml [54, 90]. Also, as the half-life of PCT is reported to be 24 hours [90], serum PCT levels are maintained at a low baseline level. Thus, there are multiple possibilities for elevated PCT levels in our population, which include but are not limited to acute kidney injury and acute liver damage. Many COVID-19 patients have

been reported to develop organ dysfunction, which includes liver [91, 92], cardiac [93, 94], and kidney injuries [95–97]. Moreover, Wang *et al*. did correlate the PCT level with acute kidney failure in COVID-19 patients [74], suggesting elevated levels of PCT in patients with severe symptoms may be related to the possible development of multiorgan failures.

It should also be noted that other health conditions also cause an increase or decrease in serum PCT values. For example, a single higher PCT value also may not necessarily indicate the presence of bacterial infection or multiorgan failure. It has been reported that with certain medications, serum PCT levels are elevated [45, 98–100]. Although these comorbidities may result in higher baseline PCT levels, obtaining a set of serial longitudinal measurements could better predict the severity trajectory (S1 Fig, in patients 1 and 4 who recovered from the disease, serial PCT values showed lower but consistent baseline values (Patient 4) or a decreasing sequential reading with time (Patient 1)). This observation warrants further prospective studies to understand the predictive symptom severity values of PCT.

It must be emphasized that the interpretation must be made in the context of other relevant lab tests, such as WBC count, IL 6, and CRP, as well as patient medical history and other pertinent information. The clinical relevance of specific PCT cut-offs in COVID-19 patient populations to predict severity (risk score) may vary from study to study (often 0.1–0.3 ng/ml) [30, 35, 38, 61, 73, 74, 101]. In this study, the value of 0.2ng/ml was used to show the difference in survival probability (Fig 4). Clearly, there is a statistically significant probability of survival with a lower serum PCT level. The rationale for using the cut-off value of 0.2ng/ml is due to the make-up of our cohort, which has many pre-existing comorbidities (Table 1), such as HTN, liver disease, diabetes, and CKD. Especially for those with CKD, higher than normal baseline PCT levels (~0.44 ng/ml) are reported regardless of whether they are on renal replacement therapy or not [102]. Even in the absence of infection, some studies report that ~36% of CKD patients have PCT levels ≥0.5 ng/ml, but in the presence of infection, ~100% would have elevated PCT ≥0.5 ng/ml. This evidence indicates that patients need to be their own controls and emphasize the importance of longitudinal serial measurements [103, 104], as seen in the box plots (Fig 2).

Taken together, the literature supports that an elevated serum PCT is expected in worsening COVID-19 cases regardless of the underlying cause. It has been reported that elevated levels of PCT in serum can be detectable within 3–4 hours following a bacterial infection [92] or other causative reasons, which highlights the possible prognostic value of PCT in COVID-19 treatment effectiveness. However, further prospective clinical studies are needed for validation.

The limitations of this study are three-fold. Firstly, to fully illuminate the association of PCT with COVID-19 disease severity, we need a controlled prospective clinical trial. It is worth noting that our study sample only included patients hospitalized with COVID-19 whose healthcare providers ordered PCT tests, in which we demonstrated that serial measurements of PCT were associated with subsequent mechanical ventilation and death. Strictly speaking, the observed associations we report are only supported for that specific subgroup, perhaps critically ill, of hospitalized COVID-19 patients whose providers were concerned enough to order PCT tests. Second, in the absence of culture data, it was difficult to rule out co-bacterial infection as the reason for the elevated PCT level. It is also possible that lack of or inadequate antibacterial treatment of such co-infection may have resulted in severe symptoms, overall outcomes, and the continuously elevated presence of serum PCT. Again, these questions can only be answered in well-designed prospective clinical trials. Third, only 10,813 patients had the longitudinal PCT values recorded within our data set. Given trends of PCT levels with COVID-19 symptom trajectory, future analyses could benefit from serial serum PCT during hospitalization. Regardless, the importance of the elevated levels of PCT in COVID-19 disease severity should not be overlooked.

## Conclusion

Frequently, COVID-19-positive patients encounter common symptoms, such as cough, shortness of breath, and fever, as well as other viral and bacterial co-infections, multiorgan failure, thrombosis, and acute kidney, liver, and heart injuries. Many survive, fending off the disease, while others succumb to death. In order to provide optimal care for these patient populations, one must have the tools to diagnose worsening symptoms early. This care should not only include identifying whether or not these patients have bacterial co-infection or need antibiotic therapy but also should include other markers predicting healthy organ functions. The data presented here showed an over 2-fold increased risk for MV and a 2-fold increased risk for in-hospital death with higher serum PCT levels. Further study might illuminate what PCT value could be considered as a prognostic indicator of symptom severity and how it might lead to better management recommendations.

## Supporting information

**S1 Table. Selected laboratory blood biochemical results of 27,154 US veterans with COVID-19 positivity and treated within the nationwide VA hospital systems.** (DOCX)

**S2 Table. Hazard ratios (HR) for subgroups of decreasing PCT values with disease progression to MV or death in veterans with multiple recorded PCT values during the hospital stay.** This analysis was performed on 10,813 veterans with at least two serum PCT tests, where a binary serum PCT change variable between the first and last serum PCT tests was created, scored as 0 if decreased (positive result), versus 1 if increased or no change (negative result). 1-unadjusted. Model 2 adjusted for demographic and comorbidity conditions (i.e., adjusted for age, race, BMI, history of CKD, Heart failure, hypertension, liver disease, and diabetes), Model 3 adjusted for demography, comorbidity conditions (Model 2) as well as pneumonia (presence or absence)and antibiotic use (used or not used). Model 4 adjusted for all covariates of Model 3 and lab test results (CRP, WBC, and lactate). (DOCX)

**S1 Fig. A set of scattergrams showing changes in serum PCT values with disease progression or the resolution of four hospitalized veterans.** While patients 1 and 4 recovered from the disease, patients 2–3 suffered in-hospital death after receiving the mechanical ventilation treatment. (TIF)

## Acknowledgments

The authors would like to express their sincere gratitude to all those who contributed to the setup and maintenance of the Veterans Affairs COVID-19 Shared Data Resource domain.

## Author Contributions

**Conceptualization:** Sujee Jeyapalina, Guo Wei, Gregory J. Stoddard, Jayant P. Agarwal.

**Data curation:** Guo Wei, Gregory J. Stoddard.

**Formal analysis:** Guo Wei, Gregory J. Stoddard.

**Investigation:** Sujee Jeyapalina, Gregory J. Stoddard, Margaret Lundquist, Merodean Huntsman, Jayant P. Agarwal.

**Methodology:** Sujee Jeyapalina, Jayant P. Agarwal.

**Resources:** Merodean Huntsman.

**Supervision:** Sujee Jeyapalina, Jayant P. Agarwal.

**Validation:** Sujee Jeyapalina.

**Writing – original draft:** Jack D. Sudduth, Margaret Lundquist, Jessica L. Marquez.

**Writing – review & editing:** Sujee Jeyapalina, Guo Wei, Gregory J. Stoddard, Jack D. Sudduth, Margaret Lundquist, Merodean Huntsman, Jessica L. Marquez, Jayant P. Agarwal.

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
