## [Decision Letter · Decision Letter 0]

27 Dec 2022

PONE-D-22-29194

Serum Procalcitonin Level is Independently Associated with Mechanical Ventilation and Case-Fatality in COVID-19 Positive US Veterans – A Potential Marker for Disease Symptom Severity and Treatment Effectiveness

PLOS ONE

Dear Dr. Jeyapalina,

Thank you for submitting your manuscript to PLOS ONE. After careful consideration, we feel that it has merit but does not fully meet PLOS ONE’s publication criteria as it currently stands. Therefore, we invite you to submit a revised version of the manuscript that addresses the points raised during the review process.

We look forward to receiving your revised manuscript.

Kind regards,

Benjamin M. Liu, MBBS, PhD, D(ABMM), MB(ASCP)

Academic Editor

PLOS ONE

Journal Requirements:

2. Please ensure that you have specified (1) whether consent was informed and (2) what type you obtained (for instance, written or verbal, and if verbal, how it was documented and witnessed). If your study included minors, state whether you obtained consent from parents or guardians. If the need for consent was waived by the ethics committee, please include this information.

3. In the ethics statement in the manuscript and in the online submission form, please provide additional information about the patient records/samples used in your retrospective study. Specifically, please ensure that you have discussed whether all data/samples were fully anonymized before you accessed them and/or whether the IRB or ethics committee waived the requirement for informed consent. If patients provided informed written consent to have data/samples from their medical records used in research, please include this information.

4. "Thank you for stating the following financial disclosure: 

   "The authors would like to express their sincere gratitude to all those who contributed to the setup and maintenance of the Veterans Affairs COVID-19 Shared Data Resource domain. This investigation was supported by an unrestricted investigator-initiated research grant from Gilead Sciences (# CO-US-983-6072). Research was also, in part, supported by the National Center for Advancing Translational Sciences of the National Institutes of Health under Award Number UL1TR002538. The content is solely the responsibility of the authors and does not necessarily represent the official views of the National Institutes of Health or Gilead Scientific Inc."

  "The authors would like to express their sincere gratitude to all those who contributed to the setup and maintenance of the Veterans Affairs COVID-19 Shared Data Resource domain. This investigation was supported by an unrestricted investigator-initiated research grant from Gilead Sciences (# CO-US-983-6072). Research was also, in part, supported by the National Center for Advancing Translational Sciences of the National Institutes of Health under Award Number UL1TR002538. The content is solely the responsibility of the authors and does not necessarily represent the official views of the National Institutes of Health or Gilead Scientific Inc"

 "The authors would like to express their sincere gratitude to all those who contributed to the setup and maintenance of the Veterans Affairs COVID-19 Shared Data Resource domain. This investigation was supported by an unrestricted investigator-initiated research grant from Gilead Sciences (# CO-US-983-6072). Research was also, in part, supported by the National Center for Advancing Translational Sciences of the National Institutes of Health under Award Number UL1TR002538. The content is solely the responsibility of the authors and does not necessarily represent the official views of the National Institutes of Health or Gilead Scientific Inc."

6. Please include your tables as part of your main manuscript and remove the individual files. Please note that supplementary tables (should remain/ be uploaded) as separate "supporting information" files

Reviewers' comments:

Reviewer's Responses to Questions

**Comments to the Author**

1. Is the manuscript technically sound, and do the data support the conclusions?

Reviewer #1: No

2. Has the statistical analysis been performed appropriately and rigorously? 

Reviewer #1: Yes

3. Have the authors made all data underlying the findings in their manuscript fully available?

Reviewer #1: Yes

4. Is the manuscript presented in an intelligible fashion and written in standard English?

Reviewer #1: Yes

5. Review Comments to the Author

Reviewer #1: In this manuscript, the authors analyzed the link between PCT and COVID. Some concerns are listed below:

1. Fig. 1, right panel, error bar is very high for last test in in-hospital mortality group.

2. In this study, in-patients were analyzed, with out-patient, non-COVID patients and healthy control missed, which will make this study based for critically ill cases.

3. PCT has some role in diagnosis of patients with sepsis. So the specificity of PCT for COVID is a concern. ROC curve analysis is warranted to set up a cutoff and determine corresponding sensitivity and specificity.

6. PLOS authors have the option to publish the peer review history of their article (what does this mean?). If published, this will include your full peer review and any attached files.

Reviewer #1: No

---

## [Author Response · Author response to Decision Letter 0]

17 Jan 2023

General 

Funding statement: Per the request we changed the funding statement as requested (see below).

“This investigation was supported by an unrestricted investigator-initiated research grant from Gilead Sciences (# CO-US-983-6072). The research was also, in part, supported by the National Center for Advancing Translational Sciences of the National Institutes of Health under Award Number UL1TR002538. The content is solely the responsibility of the authors and does not necessarily represent the official views of the National Institutes of Health or Gilead Scientific Inc. The funders had no role in study design, data collection, and analysis, decision to publish, or preparation of the manuscript”.

Tables, Figures, References, and Caption

As requested, the Tables are included within the text, and Figure Captions are inserted where they were mentioned first. The reference style is corrected to the PLOS one format (end note style downloaded from the PLOS one website).

Reviewer’s comment

General concern: Reviewer 1 marked “no” to the journal’s question 1 (is the manuscript technically sound, and do the data support the conclusion” that manuscript data?). 

Response: We have realized that our hypothesis was not clearly written in the introduction. Thus, we reworded it on Page 4, Lines 110-116. It now reads, 

“This study was therefore designed to test the primary hypothesis that there would be an association between elevated serum PCT values and symptom severity in a larger cohort of hospitalized Veterans with COVID-19 who were admitted to the nationwide Veterans Health Administration hospital system. As the secondary goals, we seek to clarify whether or not serial PCT tests would predict deteriorating clinical status in hospitalized veterans with COVID-19 positivity, as well as, whether or not the PCT levels and symptom severity disproportionally affect the Black race over the White race. If the association is replicated in this study, it would further support the pertinent benefit of using PCT to guide the prognosis of worsening COVID-19 disease.” 

Reviewer 1 concern 1: Fig. 1, right panel, the error bar is very high for the last test in the in-hospital mortality group.

Response: We agree with the reviewer that a very wide error bar, when the error bar represents a standard error or a 95% confidence interval, can be a concern as it suggests the estimate of the statistic being displayed has low precision. It appears that the reviewer glanced at the graph and misinterpreted the lines as error bars. Actually, the graph is a boxplot, where individual observations are being presented, so it is a graph showing the distribution of individual observations, being an alternative to showing a histogram. The line that the reviewer misinterpreted as an error bar is just a way to show what the maximum individual observation is. It is perfectly fine for it to be a large (wide) value, so there is actually no concern at all. We describe what the lines represent in the footnote of the graph, so readers unfamiliar with boxplots interpret them correctly.

Reviewer 1 concern 2: In this study, in-patients were analyzed, with out-patient, non-COVID patients and healthy control missed, which will make this study based on critically ill cases.

Response: We agree with the reviewer that observed associations are specific to the patient population studied, and any inference beyond that is an extrapolation, although perhaps reasonable. Ours was a very specific patient population, as the reviewer noted. We added the following sentence to the limitations in the Discussion section (Page 19, lines 445-455)

“It is worth noting that our study sample only included patients hospitalized with COVID-19 whose healthcare providers ordered PCT tests, in which we demonstrated that serial measurements of PCT were associated with subsequent mechanical ventilation and death. Strictly speaking, the observed associations we report are only supported for that specific subgroup, perhaps critically ill, of hospitalized COVID-19 patients whose providers were concerned enough to order PCT tests.”

Also, per the critique, to emphasize that this data was from critically ill patients, we also included hospitalized veterans in the title:

New title: ‘Serum Procalcitonin Level is Independently Associated with Mechanical Ventilation and Case-Fatality in Hospitalized COVID-19-Positive US Veterans – A Potential Marker for Disease Symptom Severity.” 

Reviewer 1 concern 3:

PCT has some role in the diagnosis of patients with sepsis. So the specificity of PCT for COVID is a concern. ROC curve analysis is warranted to set up a cutoff and determine corresponding sensitivity and specificity.

Response: We agree with the reviewer. We have made no attempt to determine if PCT is detecting COVID, a comorbid infection, both, or sepsis from either of both of these. A ROC curve analysis is a tool that could help answer this question, either in a diagnostic sense or in the prognostic prediction of the two outcomes we studied. It would be a big effort to do that with the needed thoroughness. We could do a quick ROC analysis, but reporting it could be misleading and so irresponsible. All our study has done is demonstrate that PCT is a candidate variable for further development into a prognostic model. Developing a PCT tool, which would include a prediction score or PCT cutoff, is something that we have already invited others to do in our manuscript. It is common to first publish papers to identify “risk factors” which provide suggestions for others to develop diagnostic or prognostic models using those factors, so our paper is still useful, only taking it as far as we have.

---

## [Decision Letter · Decision Letter 1]

6 Mar 2023

PONE-D-22-29194R1Serum Procalcitonin Level is Independently Associated with Mechanical Ventilation and Case-Fatality in Hospitalized COVID-19-Positive US Veterans – A Potential Marker for Disease SeverityPLOS ONE

Dear Dr. Jeyapalina,

Thank you for submitting your manuscript to PLOS ONE. After careful consideration, we feel that it has merit but does not fully meet PLOS ONE’s publication criteria as it currently stands. Therefore, we invite you to submit a revised version of the manuscript that addresses the points raised during the review process.

We look forward to receiving your revised manuscript.

Kind regards,

Benjamin M. Liu, MBBS, PhD, D(ABMM), MB(ASCP)

Academic Editor

PLOS ONE

Journal Requirements:

Reviewers' comments:

Reviewer's Responses to Questions

**Comments to the Author**

1. If the authors have adequately addressed your comments raised in a previous round of review and you feel that this manuscript is now acceptable for publication, you may indicate that here to bypass the “Comments to the Author” section, enter your conflict of interest statement in the “Confidential to Editor” section, and submit your "Accept" recommendation.

Reviewer #1: (No Response)

Reviewer #2: All comments have been addressed

Reviewer #3: (No Response)

Reviewer #4: All comments have been addressed

2. Is the manuscript technically sound, and do the data support the conclusions?

Reviewer #1: No

Reviewer #2: Yes

Reviewer #3: Yes

Reviewer #4: Yes

3. Has the statistical analysis been performed appropriately and rigorously? 

Reviewer #1: No

Reviewer #2: Yes

Reviewer #3: Yes

Reviewer #4: Yes

4. Have the authors made all data underlying the findings in their manuscript fully available?

Reviewer #1: Yes

Reviewer #2: Yes

Reviewer #3: Yes

Reviewer #4: Yes

5. Is the manuscript presented in an intelligible fashion and written in standard English?

Reviewer #1: Yes

Reviewer #2: Yes

Reviewer #3: Yes

Reviewer #4: Yes

6. Review Comments to the Author

Reviewer #1: While the authors addressed the first two concerns in the previous round of review, the third one, ie. the specificity of PCT for COVID prognosis is still not addressed

Reviewer #2: The paper by Jeyapalina et al. evaluated the relationship between elevated PCT levels and severe COVID-19 disease in a large VA cohort. Their study demonstrated that a high PCT (>/=0.2) is positively associated with COVID19 progression to MV or in hospital mortality. Overall, this is a well written paper, and the authors addressed the reviewer's comments appropriately.

Additional modifications are recommended:

Abstract:

- Line 25 include the most recent COVID-19 deaths (currently 6.8 million)

Introduction:

- Line 52-53, appropriately cite the WHO website

Results:

- Line 180-181, consider including PCT levels of all groups.

- Table 1, would recommend changing the table to a flow chart.

- Table 2: Age group (unknown) I am not sure what unknown refers too as it doesn't add up to the (n) of each group. Additionally, I think it would be more informative/ clinically meaningful to report the percentages per group rather than per variable. For example, % of >/= 0.1 PCT is (38.1% in the recovered without MV vs. 21.3% in the recovered after MV... etc). Would do this to all the variables. Also, consider adding %s for all hospitalized patients in all the tables.

- Table 3: Add the number of patients in each group who had repeated PCT. Also would recommend changing the way %s are reported (similar to the above comment)

Reviewer #3: The Manuscript is well organized, and technically sound. I would appreciate the authors but i have some suggestion to improve the manuscript.

1. Revise Abstract, Background portion in the abstract is quite lengthy.

2. Concise lines 63-67

3. Statement in lines 71-73 is ambiguous because not only CRP but other biomarkers are also responsible like D-dimer.

4. In the 284 replace THE WORD “covid” COVID-19

5. Most of the statements in discussion should briefly be discussed, it is quite lengthy for readers

Reviewer #4: Kindly comply with review comments. Excellent work done. COVID infection is the health alarming disease in the current world. need further suggestion and research work to help in understanding the exact mechanism and pathogenesis of the COVID disease.

7. PLOS authors have the option to publish the peer review history of their article (what does this mean?). If published, this will include your full peer review and any attached files.

Reviewer #1: No

Reviewer #2: No

Reviewer #3: No

Reviewer #4: **Yes: **Muhammad Taj Akbar

---

## [Author Response · Author response to Decision Letter 1]

20 Mar 2023

Thank you for the feedback. We have reviewed the comments received and made the appropriate changes. There are given below.

1. Reviewer #1 Concern 1: While the authors addressed the first two concerns in the previous round of review, the third one, i.e., the specificity of PCT for COVID prognosis, is still not addressed.

Response: Combining this concern with how Reviewer 1 stated it previously, the reviewer is specifically referring to how accurately, based on a receiver operating characteristic (ROC) analysis, PCT predicts our outcomes in COVID patients. Firstly, the aim of this study was to understand the association of PCT with poor survival outcomes. We did not aim to understand the prognostic value of the PCT. Thus, we did not attempt to do the ROC analysis previously. We did convey this message in the comment. However, it appears our explanation was not satisfactory to Reviewer 1. Thus, as requested by Reviewer 1, we performed the ROC analysis, which is given below, as we felt Reviewer 1 is curious to see it. Achieving a ROC of 0.70 would require a multivariable prediction equation that includes PCT, where the equation is derived from a prospective study with PCT testing being randomized. However, the ROC area was lower (0.53-0.60).We did not feel below ROC analysis would add any further value to the manuscript and hence did not include it.

A cursory evaluation of PCT as a prognostic indicator of ventilation and death: The aim of this study was not to derive a prognostic model of ventilation and death in COVID-19 patients. The results presented, however, suggest that PCT is a useful prognostic indicator of these outcomes. To support that suggestion, we used PCT by itself as the only predictor variable to obtain prognostic model test characteristics. We first limited the data to the n=10,813 patient cohort with at least two PCT laboratory values preceding ventilation, death, or discharge, whichever came first. We then used the first PCT value to predict ventilation and death. We report the area under the ROC curve when PCT is used as a continuous variable. Then, we formed a binary predictor by dichotomizing the PCT variable at the optimal cutpoint, which was the value of PCT at the point on the ROC curve closest to the upper left corner (where sensitivity=0 and 1-specificity=0). Finally, we show the test characteristics (ROC area, sensitivity, and specificity) for the binary predictor. We repeated this using the second PCT value to see if test characteristics improved, which would support the usefulness of serial measurements of PCT (Refer to the Table given in response to reviewer letter).

Although we provided some PCT cutpoints (< cutpoint = outcome likely not to occur ; ≥ cutpoint = outcome likely to occur) to predict ventilation or death during the hospitalization, this was only to give a sense of how predictive it is. We do not recommend these cutpoints be used in clinical practice for several reasons: the cutpoints were derived in the limited situation when an PCT test is ordered for clinical reasons and thus only apply to patients in that specific situation, the cutpoints have not been validated, the ROC area did not meet the 0.70 threshold for being a clinically useful test, and a proper prognostic model should include other variables and be derived in a prospective study.

2. Reviewer 2 concern 1: “Line 25 - include the most recent COVID-19 deaths (currently 6.8 million)”

 Response: Changed as suggested (current Lines 26 and 52)

3. Reviewer 2 concern 2: “Line 52-53, appropriately cite the WHO website”

Response: The WHO website is included as the reference [4], and all other references thereafter were renumbered.

4. Reviewer 2 Result section:

• Line 180-181, consider including PCT levels of all groups.

• Table 1, would recommend changing the table to a flow chart.

• Table 2: Age group (unknown) I am not sure what unknown refers to as it doesn't add up to the (n) of each group. Additionally, I think it would be more informative/ clinically meaningful to report the percentages per group rather than per variable. For example, % of >/= 0.1 PCT is (38.1% in the recovered without MV vs. 21.3% in the recovered after MV... etc). Would do this to all the variables. Also, consider adding %s for all hospitalized patients in all the tables.

• Table 3: Add the number of patients in each group who had repeated PCT. Also would recommend changing the way %s are reported (similar to the above comment)

Response: All of the above suggestions are now incorporated into the manuscript. 

• Written as “This value was 3 times higher in veterans who suffered in-hospital death (0.3 (0.1-0.8) ng/ml) after receiving mechanical ventilation and 2 times higher for those who recovered after ventilation or those who perished without ventilation (0.2 (0.1-0.6) ng/ml).” Now Lines 176-178”.

• Table 1 is now redone into a Figure (Fig 1)

• Previous Table 2 is now renamed as Table 1. An explanation of “the unknown” is given in the caption.

• As suggested, the percentages of the group are given in previous Tables 2-3 (new Tables 3-4). 

5. Reviewer 3 concern 1: Revise Abstract, Background portion in the abstract is quite lengthy.

Response: As suggested, the background section is now revised in the Abstract. Lines 26-30. 

6. Reviewer 3 concern 2: Concise lines 63-67

Response: Previous lines 63-67 revised to read (now Lines 64-65), “Amongst them, serum level markers, as well as immune cell counts, have all been associated with symptom severities, including case-fatalities [14-26].”

7. Reviewer 3 concern 3: The statement in lines 71-73 is ambiguous because not only CRP but other biomarkers are also responsible, like D-dimer.

Response: Previous lines 71-73 revised to read, “Previous COVID-19 studies have suggested that the blood plasma laboratory values of PCT as well as others such as CRP, D-dimer, etc.,…” – (new lines 70-71)

8. Reviewer 3 concern 4: In the 284, replace THE WORD “covid” COVID-19.

Response: Revised as suggested.

9. Reviewer 3 concern 5: Most of the statements in the discussion should briefly be discussed; it is quite lengthy for readers:

Response: Revised as suggested. The race analysis section did not add any value to the manuscript. Hence, we removed race references from all sections to make the manuscript more concise and relevant.

---

## [Decision Letter · Decision Letter 2]

3 Apr 2023

Serum Procalcitonin Level is Independently Associated with Mechanical Ventilation and Case-Fatality in Hospitalized COVID-19-Positive US Veterans – A Potential Marker for Disease Severity

PONE-D-22-29194R2

Dear Dr. Jeyapalina,

We’re pleased to inform you that your manuscript has been judged scientifically suitable for publication and will be formally accepted for publication once it meets all outstanding technical requirements.

Kind regards,

Benjamin M. Liu, MBBS, PhD, D(ABMM), MB(ASCP)

Academic Editor

PLOS ONE

Additional Editor Comments (optional):

Reviewers' comments:

Reviewer's Responses to Questions

**Comments to the Author**

1. If the authors have adequately addressed your comments raised in a previous round of review and you feel that this manuscript is now acceptable for publication, you may indicate that here to bypass the “Comments to the Author” section, enter your conflict of interest statement in the “Confidential to Editor” section, and submit your "Accept" recommendation.

Reviewer #2: All comments have been addressed

Reviewer #3: All comments have been addressed

2. Is the manuscript technically sound, and do the data support the conclusions?

Reviewer #2: Yes

Reviewer #3: Yes

3. Has the statistical analysis been performed appropriately and rigorously? 

Reviewer #2: Yes

Reviewer #3: Yes

4. Have the authors made all data underlying the findings in their manuscript fully available?

Reviewer #2: Yes

Reviewer #3: Yes

5. Is the manuscript presented in an intelligible fashion and written in standard English?

Reviewer #2: Yes

Reviewer #3: No

6. Review Comments to the Author

Reviewer #2: The authors have nicely addressed all my comments and concerns. No additional comments and I suggest publication.

Reviewer #3: (No Response)

7. PLOS authors have the option to publish the peer review history of their article (what does this mean?). If published, this will include your full peer review and any attached files.

Reviewer #2: No

Reviewer #3: No

---

## [Editor Report · Acceptance letter]

6 Apr 2023

PONE-D-22-29194R2 

Serum Procalcitonin Level is Independently Associated with Mechanical Ventilation and Case-Fatality in Hospitalized COVID-19-Positive US Veterans – A Potential Marker for Disease Severity 

Dear Dr. Jeyapalina:

I'm pleased to inform you that your manuscript has been deemed suitable for publication in PLOS ONE. Congratulations! Your manuscript is now with our production department. 

Kind regards, 

on behalf of

Dr. Benjamin M. Liu 

Academic Editor

PLOS ONE